# An Iterative Method Based on the Marginalized Particle Filter for Nonlinear B-Spline Data Approximation and Trajectory Optimization

**Jens Jauch** **, Felix Bleimund, Michael Frey \*** **and Frank Gauterin**

Institute of Vehicle System Technology, Karlsruhe Institute of Technology, 76131 Karlsruhe, Germany; jens.jauch@kit.edu (J.J.); f.blmnd@posteo.de (F.B.); frank.gauterin@kit.edu (F.G.)
*   Correspondence: michael.frey@kit.edu

**Abstract:** The B-spline function representation is commonly used for data approximation and trajectory definition, but filter-based methods for nonlinear weighted least squares (NWLS) approximation are restricted to a bounded definition range. We present an algorithm termed nonlinear recursive B-spline approximation (NRBA) for an iterative NWLS approximation of an unbounded set of data points by a B-spline function. NRBA is based on a marginalized particle filter (MPF), in which a Kalman filter (KF) solves the linear subproblem optimally while a particle filter (PF) deals with nonlinear approximation goals. NRBA can adjust the bounded definition range of the approximating B-spline function during run-time such that, regardless of the initially chosen definition range, all data points can be processed. In numerical experiments, NRBA achieves approximation results close to those of the Levenberg–Marquardt algorithm. An NWLS approximation problem is a nonlinear optimization problem. The direct trajectory optimization approach also leads to a nonlinear problem. The computational effort of most solution methods grows exponentially with the trajectory length. We demonstrate how NRBA can be applied for a multiobjective trajectory optimization for a battery electric vehicle in order to determine an energy-efficient velocity trajectory. With NRBA, the effort increases only linearly with the processed data points and the trajectory length.

**Keywords:** nonlinear; recursive; iterative; B-spline; approximation; marginalized particle filter; Rao-Blackwellized particle filter; multiobjective; trajectory; optimization

## 1. Introduction

B-spline functions, curves, and surfaces are widely used for approximation [1–3] and for defining the trajectories of vehicles [4,5], robots [6,7] and industrial machines [8]. Furthermore, they are common in computer graphics [9,10] and signal processing for filter design and signal representation [11–15].

We address the approximation of a set of data points by a B-spline function in the nonlinear weighted least squares (NWLS) sense as well as the nonlinear optimization of a B-spline trajectory. In both cases, a Bayesian filter determines the coefficients of the B-spline function.

### 1.1. Nonlinear Weighted Least Squares Data Approximation

In NWLS approximation problems, the solution depends on the function coefficients in a nonlinear fashion. Based on the results of numerical experiments, Reference [16] reported that a B-spline function was beneficial in solving NWLS problems because of its piecewise polynomial character and smoothness.

In offline applications, a bounded number of data points needs to be processed and all data points are known at the same time. Therefore, the problem can be solved using a batch method. Batch methods

for NWLS problems include the Gauss–Newton algorithm and the Levenberg-Marquardt (LM) algorithm. None of these algorithms is an exact method [16]. The LM algorithm solves in each iteration a linearized NWLS problem [17]. A method for separable NWLS problems, in which some parameters affect the solution linearly, is derived in References [18,19].

In contrast, in online applications such as signal processing, data points become available consecutively and their number is often unbounded. Sliding window algorithms keep the required memory constant by processing only a subset consisting of the latest data points [20]. A sliding window implementation of the LM algorithm for online applications is proposed in [21].

Recursive methods only store an already existing solution and update it with each additional data point. Therefore, they are suitable for online applications and usually require less memory and computational effort than batch algorithms that have been adapted for online applications.

NWLS approximation problems are nonlinear optimization problems. Therefore, recursive algorithms for NWLS problems can be based on nonlinear Bayesian filters.

*1.2. Trajectory Optimization*

Many driver assistance systems calculate a desired vehicle movement, also denoted trajectory, by solving a multiobjective optimization problem with respect to target criteria such as comfort, safety, energy consumption, and travel time. The trajectory optimization methods can be divided into Dynamic Programming (DP), direct methods (DM), and indirect methods (IM).

DP is based on Bellmann's principle of optimality and determines globally optimal solutions. Its computational effort grows linearly with the temporal length of the trajectory and exponentially with the dimensions of the optimization problem. An adaptive cruise control based on DP is proposed in Reference [22]. DP-based algorithms for energy-efficient automated vehicle longitudinal control exist for vehicles with an internal combustion engine [23], hybrid electric vehicles [24], and plug-in hybrid electric vehicles [25]. In vehicles with a conventional powertrain, one dimension of the optimization problem refers to the selected gear. In case of a vehicle with a hybrid powertrain, there is at least one additional dimension for the operating mode, i.e., how power flows between the combustion engine, electric motor, and wheels. These degrees of freedom come along with various constraints, and frequently, the optimization problem needs to be simplified such that it can be solved in real-time.

DM lead to an optimization problem, in which the optimization variables are the parameters of a functional trajectory representation. The problem is usually nonlinear and solved using sequential quadratic programming methods or interior point methods. An example for a DM is the model predictive control, which solves the trajectory optimization problem on a receding horizon. DM are locally optimal, and their computational effort grows polynomially with the dimensions but mostly exponentially with the temporal trajectory length. Therefore, the optimization horizon is usually restricted to a few seconds.

IM are based on variational calculus and require solving a nonlinear equation system. They offer a polynomial complexity increase with the number of dimensions and the time horizon.

In practice, mainly the two first approaches are used and combined for solving difficult, farsighted trajectory optimization problems because of their complementary properties. Then DP provides a rough long-term reference trajectory for a DM that computes feasible trajectories within a short horizon [26,27].

*1.3. Bayesian Filters*

The Bayesian approach to a state estimation for dynamic systems calculates the probability density function (pdf) of the unknown system state. The required information stems partly from a system model and partly from previous measurements. The state estimation is performed by a recursive filter that alternates between a time update that predicts the state via the system model and a measurement update that corrects the estimate with the current measurement.

The Kalman filter (KF) computes an optimal state estimate for systems with linear system and measurement equations and Gaussian system and measurement noises [28]. Use cases include path planning applications [29]. However, in many scenarios, the linear Gaussian assumptions do not apply and suboptimal approximate nonlinear Bayesian filters such as the extended Kalman filter (EKF), unscented Kalman filter (UKF), or particle filter (PF) are required [30].

The EKF applies a local first order Taylor approximation to the nonlinear system and measurement functions via Jacobians in order to keep the linear state and measurement equations. The system and measurement noises are both approximated with Gaussian pdfs [28]. Although the EKF is not suitable for systems with a strong nonlinearity or non-Gaussian noise, it is still often successfully used for a nonlinear state estimation [31]. For example, an NWLS approximation via a modified EKF is presented in Reference [32].

An alternative to the approximation of the nonlinear state and measurement functions is the approximation of the pdfs. This can be done by propagating a few state samples called sigma points through the nonlinear functions. A filter that follows this approach is referred to as a sigma point Kalman filter. One of the most well-known representatives is the UKF. It uses $2 \cdot J + 1$ deterministically chosen sigma points, whereby $J$ denotes the dimensions of the system state. The pdfs are approximated as Gaussians of which the means and variances are determined from the propagated sigma points [28].

Compared to the EKF, the UKF offers at least a second-order accuracy [33] and is a derivative-free filter [28], meaning that it does not require the evaluation of Jacobians, which is often computationally expensive in the EKF [31]. Several publications report nonlinear problems in which the UKF performs better than the EKF, e.g., for a trajectory estimation [33,34]. However, if the pdf cannot be well-approximated by a Gaussian because the pdf is multimodal or has a strong skew, the UKF will also not perform well. Under such conditions, sequential Monte Carlo methods like the PF outperform Gaussian filters like EKF and UKF [30].

The PF approximates the pdf by a large set of randomly chosen state samples called particles. The state estimate is a weighted average of the particles. With increasing number of particles, the pdf approximation by the particles becomes equivalent to the functional pdf representation and the estimate converges against the optimal estimate [30]. For nonlinear and non-Gaussian systems, the PF allows the determination of various statistical moments, whereas EKF and UKF are limited to the approximation of the first two moments [31]. However, the number of particles that is needed for a sufficient approximation of the pdf increases exponentially with the state dimension [35]. The PF has been applied to the optimization [36] and prediction [37] of trajectories successfully as well.

Many use cases involve a mixed linear/nonlinear system. Typically, there are few nonlinear state dimensions and comparatively many linear Gaussian state dimensions. The marginalized particle filter (MPF) is beneficial for such problems as it combines KF and PF. The PF is only applied to the nonlinear states because the linear part of the state vector is marginalized out and optimally filtered with the KF. This approach is known as Rao–Blackwellization and can be described as an optimal Gaussian mixture approximation. Therefore, the MPF is also called a Rao–Blackwellized particle filter or mixture Kalman filter. Marginalizing out linear states from the PF strongly reduces the computational effort because less particles suffice and often enables real-time applications. Simultaneously, the estimation accuracy usually increases [31,38].

In the recent past, several publications have proposed approaches for localization [39,40] and trajectory tracking [38,41] that are based on the MPF because of its advantages for mixed linear/nonlinear systems. Automotive use cases include a road target tracking application, of which the multimodality requires using a PF or MPF [42]. The MPF is chosen as it allows a reduction in the number of particles for less computational effort. Similarly, Reference [35] presents a MPF application for lane tracking, in which the achieved particle reduction compared to a pure PF enables the execution of the algorithm in real-time in an embedded system.

### *1.4. Contribution*

By definition, a B-spline function with a bounded number of coefficients has a bounded definition range. Usually, approximation algorithms require a bounded number of coefficients which restricts the approximation of data points with a B-spline function to a bounded interval that needs to be determined in advance.

In online applications, the required B-spline function definition range might not be precisely known or vary. This can result in the issuse of unprocessable data points outside the selected definition range.

In Reference [43], we presented the recursive B-spline approximation (RBA) algorithm, which iteratively approximates an unbounded set of data points in the linear weighted least squares (WLS) sense with a B-spline function using a KF. A novel shift operation enables an adaptation of the definition range during run-time such that the latest data point can always be approximated.

However, recursive NWLS B-spline approximation methods are still restricted to a constant approximation interval. We contribute to closing this research gap by proposing and investigating an algorithm termed nonlinear recursive B-spline approximation (NRBA) for the case of NWLS approximation problems.

NRBA comprises an MPF that addresses nonlinear target criteria with its PF while it determines the optimal solution for linear target criteria with a KF [44]. The target criteria that refer to the value of the B-spline function or its derivatives directly are linear criteria. Hereby, the benefit of using MPF is that it can deal with strong nonlinearities, that its computational effort can be adapted by changing the number of particles in order to meet computation time constraints, and that it accepts the known measurement matrix for linear target critera as an input, whereas other nonlinear filters estimate the relationship between measurements and function coefficients.

In automotive applications, the exponential growth of the computational effort with an increasing time horizon limits the application of DM to short time horizons. Hence, the research gap regarding trajectory optimization consists of available DM with a lower complexity. Compared to conventional and hybrid vehicles, the powertrain of a battery electric vehicle (BEV) often only has a constant gear ratio which enables savings in computational effort.

Since the NWLS approximation problem that NRBA solves is an unconstrained nonlinear optimization problem, NRBA can be applied for multiobjective trajectory optimization. Our contribution regarding trajectory optimization is an iterative local direct optimization method for B-spline trajectories of which the computational effort only grows linearly with the time horizon instead of exponentially. Due to the iterative nature of NRBA, the optimization can be paused, and if computation time is available, the temporal length of the trajectory can be extended by calculating additional coefficients.

### *1.5. Structure of the Data Set*

Analogous to Reference [43], we consider the data point sequence $\{(s_t, \boldsymbol{y}_t)\}_{t=1,2,\dots,n}$. The index $t$ indicates the time step at which the data point $(s_t, \boldsymbol{y}_t)$ becomes available. $s_t$ denotes the value of the independent variable $s$ at $t$. The vector $\boldsymbol{y}_t = (y_{t,1}, y_{t,2}, \dots, y_{t,v}, \dots, y_{t,V_t})^\top$ summarizes $V_t$ scalar measurements $y$. The superscript $^\top$ indicates transposed quantities. $V_t \in \mathbb{N}$ can vary with $t$, but we suppose that $V_t \ll n \; \forall t$. The vector $\boldsymbol{y}$ comprises all measurements and is given by

$$\boldsymbol{y} = (\underbrace{y_{1,1}, \dots, y_{1,V_1}}_{=:\boldsymbol{y}_1^\top}, \dots, \boldsymbol{y}_t^\top, \dots, \underbrace{y_{n,1}, \dots, y_{n,V_n}}_{=:\boldsymbol{y}_n^\top})^\top \tag{1}$$

### *1.6. Outline*

Section 2.1 states the used B-spline function definition. In Section 2.2, we specify the MPF and the chosen state-space model. Section 2.3 proposes the NRBA algorithm for an NWLS approximation. The numerical experiments in Section 3 investigate the capabilities of NRBA compared to the LM algorithm as well as the influences of the NRBA parameters on the result and convergence before

we demonstrate how NRBA can be applied for a multiobjective trajectory optimization in Section 4. In Section 5, we recapitulate the features of NRBA and conclude.

## 2. Methods

### 2.1. B-Spline Function Representation

The value of a B-spline function results from the weighted sum of $J$ polynomial basis functions called B-splines. All B-splines possess the same degree $d$. The B-splines are defined by $d$, and the knot vector $\kappa = (\kappa_1, \kappa_2, \ldots, \kappa_{J+d+1})$. We suppose that the values of the knots $\kappa$ grow strictly monotonously ($\kappa_k < \kappa_{k+1}$, $k = 1, 2, \ldots, J+d$). $\mu$ with $d+1 \leq \mu \leq J$ is the spline interval index and $[\kappa_\mu, \kappa_{\mu+1})$ is the corresponding spline interval of the B-spline function.

In the $j$th B-spline $b_j(s)$, $j = 1, 2, \ldots, J$ is positive for $s \in (\kappa_j, \kappa_{j+d+1})$ and diminishes everywhere else. This feature is referred to as local support and causes the B-spline function to be piecewise defined for each spline interval. For $s \in [\kappa_\mu, \kappa_{\mu+1})$, only the B-splines $b_j(s)$, $j = \mu - d, \ldots, \mu$ can be positive.

Their values for a specific $s$ are comprised in the B-spline vector $b_{\mu,d}(s) = (b_{\mu-d}(s), b_{\mu-d+1}(s), \ldots, b_\mu(s)) \in \mathbb{R}^{1 \times (d+1)}$ which is calculated according to Equation (2):

$$b_{\mu,d}(s) = \underbrace{B_{\mu,1}(s)}_{\in \mathbb{R}^{1 \times 2}} \underbrace{B_{\mu,2}(s)}_{\in \mathbb{R}^{2 \times 3}} \ldots \underbrace{B_{\mu,\delta}(s)}_{\in \mathbb{R}^{\delta \times (\delta+1)}} \ldots \underbrace{B_{\mu,d}(s)}_{\in \mathbb{R}^{d \times (d+1)}} \tag{2}$$

The B-spline matrix $B_{\mu,\delta}(s) \in \mathbb{R}^{\delta \times (\delta+1)}$ with $\delta \in \mathbb{N}$ and $\delta \leq d$ reads

$$B_{\mu,\delta}(s) = \begin{bmatrix} \frac{\kappa_{\mu+1}-s}{\kappa_{\mu+1}-\kappa_{\mu+1-\delta}} & \frac{s-\kappa_{\mu+1-\delta}}{\kappa_{\mu+1}-\kappa_{\mu+1-\delta}} & 0 & \cdots & 0 \\ 0 & \frac{\kappa_{\mu+2}-s}{\kappa_{\mu+2}-\kappa_{\mu+2-\delta}} & \frac{s-\kappa_{\mu+2-\delta}}{\kappa_{\mu+2}-\kappa_{\mu+2-\delta}} & \cdots & 0 \\ \vdots & \vdots & \ddots & \ddots & \vdots \\ 0 & 0 & \cdots & \frac{\kappa_{\mu+\delta}-s}{\kappa_{\mu+\delta}-\kappa_\mu} & \frac{s-\kappa_\mu}{\kappa_{\mu+\delta}-\kappa_\mu} \end{bmatrix}. \tag{3}$$

$\mathcal{D} = [\kappa_{d+1}, \kappa_{J+1})$ is the definition range of the B-spline function $f : \mathcal{D} \to \mathbb{R}$, $s \mapsto f(s)$. For $s \in [\kappa_\mu, \kappa_{\mu+1})$, $f$ is defined by

$$f(s) = b_{\mu,d}(s) x_{\mu,d} \tag{4}$$

with coefficient vector

$$x_{\mu,d} = \left( x_{\mu-d}, x_{\mu-d+1}, \ldots, x_\mu \right)^\top. \tag{5}$$

$f$ has $d - 1$ continuous derivatives. For $r \in \mathbb{N}_0$, the $r$th derivative $\frac{\partial^r}{\partial s^r} f(s)$ of $f$ reads

$$\frac{\partial^r}{\partial s^r} f(s) = \frac{\partial^r}{\partial s^r} b_{\mu,d}(s) x_{\mu,d} \tag{6}$$

with B-spline vector

$$\frac{\partial^r}{\partial s^r} b_{\mu,d}(s) = \begin{cases} \frac{d!}{(d-r)!} B_{\mu,1}(s) \ldots B_{\mu,d-r}(s) B'_{\mu,d-r+1} \ldots B'_{\mu,d}, & \text{if } r \leq d \\ \mathbf{0}_{1 \times (d+1)}, & \text{otherwise.} \end{cases} \tag{7}$$

$\mathbf{0}_{1 \times (d+1)}$ is a $1 \times (d+1)$ zero matrix. The matrix $B'_{\mu,\delta} \in \mathbb{R}^{\delta \times (\delta+1)}$ results from computing the derivative with respect to $s$ for each element of $B_{\mu,\delta}(s)$ [43,45]:

$$B'_{\mu,\delta} = \begin{bmatrix} \frac{-1}{\kappa_{\mu+1}-\kappa_{\mu+1-\delta}} & \frac{1}{\kappa_{\mu+1}-\kappa_{\mu+1-\delta}} & \cdots & 0 \\ \vdots & \ddots & \ddots & \vdots \\ 0 & \cdots & \frac{-1}{\kappa_{\mu+\delta}-\kappa_\mu} & \frac{1}{\kappa_{\mu+\delta}-\kappa_\mu} \end{bmatrix} \tag{8}$$

### 2.2. Marginalized Particle Filter

The marginalized particle filter (MPF) is an iterative algorithm for estimating the unknown state vector $x_t$ of a system at time step $t \in \mathbb{N}$.

In the MPF, $x_t$ is subdivided into $x_t = \left( \left(x_t^L\right)^\top, \left(x_t^N\right)^\top \right)^\top$, whereby the KF optimally estimates the linear state vector $x_t^L$ and a PF estimates the nonlinear state vector $x_t^N$. Exploiting linear substructures allows for better estimates and a reduction of the computational effort. Therefore, the MPF is beneficial for mixed linear/nonlinear state-space models [46]. Due to Equations (4) and (6), linear substructures will occur in approximation problems as long as there are target criteria that refer to the value of the B-spline function or its derivatives directly.

MPF algorithms for several state-space models can be found in Reference [46] along with a MATLAB example that can be downloaded from [47]. An equivalent but new formulation of the MPF that allows for reused, efficient, and well-studied implementations of standard filtering components is stated in Reference [44].

For an NWLS approximation, we apply the following state-space model derived from Reference [44]:

$$x_{t+1}^N = \mathcal{A}_t^N x_t^N + \omega_t^N + u_t^N \qquad \text{(nonlinear state equation)} \tag{9}$$
$$x_{t+1}^L = \mathcal{A}_t^L x_t^L + \omega_t^L + u_t^L \qquad \text{(linear state equation)} \tag{10}$$
$$y_t = \mathcal{C} x_t^L + c\left(x_t^N\right) + v_t \qquad \text{(measurement equation)} \tag{11}$$

The superscripts $L$ and $N$ indicate that the corresponding quantity refers to linear or nonlinear state variables, respectively. $\mathcal{A}_t$ denotes the state transition matrix, $u_t$ is the known input vector, $y_t$ is the vector of measurements, $\mathcal{C}_t$ is the measurement matrix, and $c$ is the nonlinear measurement function that depends on $x_t^N$.

$\omega_t^L$ denotes the process noise of the linear state vector with a covariance matrix $\mathcal{Q}_t^L$, $\omega_t^N$ is the process noise of the nonlinear state vector with a covariance matrix $\mathcal{Q}_t^N$, and $v_t$ is the measurement noise with a covariance matrix $\mathcal{R}_t$.

The model of the conditionally linear subsystem in the KF has the state vector $\left( \xi^\top, \left(x^L\right)^\top \right)^\top$, whereby $\xi$ describes the linear dynamics of $x^N$:

$$\begin{pmatrix} \xi_{t+1} \\ x_{t+1}^L \end{pmatrix} = \begin{pmatrix} 0 & \mathcal{A}_t^N \\ 0 & \mathcal{A}_t^L \end{pmatrix} \begin{pmatrix} \xi_t \\ x_t^L \end{pmatrix} + \begin{pmatrix} u_t^N \\ u_t^L \end{pmatrix} + \begin{pmatrix} \omega_t^N \\ \omega_t^L \end{pmatrix}$$
$$y_t = \begin{pmatrix} 0 & \mathcal{C}_t \end{pmatrix} \begin{pmatrix} \xi_t \\ x_t^L \end{pmatrix} + c\left(x_t^L\right) + v_t \tag{12}$$

The covariance matrix of process noise is $\begin{pmatrix} \mathcal{Q}_t^N & 0 \\ 0 & \mathcal{Q}_t^L \end{pmatrix}$, and $0$ denotes a zero matrix with a suitable size.

A PF with the model

$$x_{t+1}^N = \bar{\omega}_t^N$$
$$y_t = \bar{v}_t \tag{13}$$

deals with the remaining nonlinear effects. The noise depends on the estimates indicated by ˆ from the conditionally linear model:

$$\bar{\omega}_t^N \sim \mathcal{N}\left( \hat{\xi}_t, \mathcal{P}_t^{\xi,-} \right)$$
$$\bar{v}_t \sim \mathcal{N}\left( c\left(x_t^N\right) + \mathcal{C}_t\left(x_t^N\right) \hat{x}^{L,-}, S_t \right) \tag{14}$$

with

$$S_t = \mathcal{C}_t \mathcal{P}_t^{L,-} \mathcal{C}_t^\top + \mathcal{R}_t \tag{15}$$

where the superscript $^-$ refers to a priori quantities that are computed in the time update which is based on the state of Equations (9) and (10). In contrast, $^+$ denotes a posteriori quantities that are calculated in the following measurement update based on the measurement of Equation (11).

$\mathcal{P}_t^{L,-}$ and $\mathcal{P}_t^{\xi,-}$ are the covariance matrices of the estimation errors that belong to $\hat{x}_t^L$ and $\hat{\xi}_t$, respectively.

The PF uses multiple state estimates called particles simultaneously. The superscript $^p$ with $p = 1, \ldots, P$ is the particle index and $P$ is the particle count. In general, a KF is used for each particle. In the chosen state-space model, however, $\mathcal{A}_t^L$, $\mathcal{A}_t^N$, $\mathcal{Q}_t^L$, and $\mathcal{Q}_t^N$ are independent of $x_t^L$ and $x_t^N$. This implies that $\mathcal{P}_t^{L,-}$ and $\mathcal{P}_t^{\xi,-}$ are identical for all KFs which reduces the computational effort substantially [44,46].

Algorithm 1 states the equations for one MPF iteration and was derived from References [44,46]. For an implementation in MATLAB, we adapted the example from Reference [47]. Note that, in Algorithm 1, the measurement update of the previous time step $t - 1$ occurs before the time update for the current time step $t$, similar to the algorithm in Reference [48].

---

**Algorithm 1:** The marginalized particle filter derived from References [44,46]

---

**Input:** $\mathcal{A}_t^L, \mathcal{A}_t^N, \mathcal{C}_{t-1}, c, \mathcal{P}_{t-1}^{L,-}, \mathcal{Q}_t^L, \mathcal{Q}_t^N, \mathcal{R}_{t-1}, u_t^L, u_t^N, \hat{x}_{t-1}^{L,-,p}, \hat{x}_{t-1}^{N,-,p}, y_{t-1}$

/* 1a) PF measurement update                      */

1   For $p = 1, \ldots, P$, compute the particle importance weights $q_t^p$ using the likelihood

$$q_t^p = \mathcal{N}(\hat{y}^p, S_t),\ \hat{y}^p = \mathcal{C}_{t-1}\mathcal{P}_{t-1}^{L,-}\hat{x}_{t-1}^{L,-,p} + c\left(\hat{x}_{t-1}^{N,-,p}\right),\ S_{t-1} = \mathcal{C}_{t-1}\mathcal{P}_{t-1}^{L,-}\mathcal{C}_{t-1}^\top + \mathcal{R}_{t-1}\ \text{and}$$

compute the normalized weights $\tilde{q}_t^p = \dfrac{q_t^p}{\sum_{p'=1}^P q_t^{(p')}}$.

/* 1b KF measurement update                      */

2   $\hat{x}_{t-1}^{L,+,p} \leftarrow \hat{x}_{t-1}^{L,-,p} + \mathcal{P}_{t-1}^{L,-}\mathcal{C}_{t-1}^\top S_{t-1}^{-1}\left(y_{t-1} - \hat{y}^p\right)$

3   $\mathcal{P}_{t-1}^{L,+} \leftarrow \mathcal{P}_{t-1}^{L,-} - \mathcal{P}_{t-1}^{L,-}\mathcal{C}_{t-1}^\top S_{t-1}^{-1}\mathcal{C}_{t-1}\mathcal{P}_{t-1}^{L,-}$

/* 1c Resampling                                  */

4   Resample $P$ particles with replacement, probability $\left(\hat{x}_{t-1}^{L,+,(p')} = \hat{x}_{t-1}^{L,+,p}\right) = \tilde{q}_t^p$.

5   $\hat{x}_{t-1}^+ \leftarrow$ mean of $\hat{x}_{t-1}^{L,+,p}$, $p = 1, \ldots, P$

/* 2a KF time update                            */

6   $\hat{x}_t^{L,-,p} \leftarrow \mathcal{A}_t^L \hat{x}_{t-1}^{L,+,p} + u_t^L$

7   $\hat{\xi}_t^p \leftarrow \mathcal{A}_t^N \hat{x}_{t-1}^{L,+,p} + u_t^N$

8   $\mathcal{P}_t^{L,-} \leftarrow \mathcal{A}_t^L \mathcal{P}_{t-1}^{L,+}\left(\mathcal{A}_t^L\right)^\top + \mathcal{Q}_t^L$

9   $\mathcal{P}_t^{\xi,-} \leftarrow \mathcal{A}_t^N \mathcal{P}_{t-1}^{L,+}\left(\mathcal{A}_t^N\right)^\top + \mathcal{Q}_t^N$

10   $\mathcal{P}_t^{\xi L,-} \leftarrow \mathcal{A}_t^N \mathcal{P}_{t-1}^{L,+}\left(\mathcal{A}_t^L\right)^\top$

11   $\mathcal{P}_t^{L\xi,-} \leftarrow \left(\mathcal{P}_t^{\xi L,-}\right)^\top$

/* 2b PF time update                            */

12   For $p = 1, \ldots, P$, predict new particles, $\hat{x}_t^{N,-,p} \sim \mathcal{N}\left(\hat{\xi}_t^p, \mathcal{P}_t^{\xi,-}\right)$.

/* 2c Mixing step, update KF                    */

13   $\hat{x}_t^{L,-,p} \leftarrow \hat{x}_t^{L,-,p} + \mathcal{P}_t^{L\xi,-}\left(\mathcal{P}_t^{\xi,-}\right)^{-1}\left(\hat{x}_t^{N,-,p} - \hat{\xi}_t^p\right)$

14   $\mathcal{P}_t^{L,-} \leftarrow \mathcal{P}_t^{L,-} - \mathcal{P}_t^{L\xi,-}\left(\mathcal{P}_t^{\xi,-}\right)^{-1}\mathcal{P}_t^{\xi L,-}$

**Output:** $\mathcal{P}_t^{L,-}, \hat{x}_{t-1}^+, \hat{x}_t^{L,-,p}, \hat{x}_t^{N,-,p}$

---

In line 4 of Algorithm 1, linear particles are resampled according to their corresponding normalized importance weights. After resampling, particles with a low measurement error occur more frequently in the set of particles. Subsequently, all particles $\hat{x}_{t-1}^{L,+,p}$ are aggregated in line 5 to a single estimate $\hat{x}_{t-1}^+$ by calculating their mean.

After both KF and PF have been time updated, the KF is adjusted based on the PF estimates in a mixing step with the cross-covariances of the estimation errors, $\mathcal{P}_t^{\xi L,-}$ and $\mathcal{P}_t^{L\xi,-}$.

In the new formulation from Reference [44], resampling occurs after the measurement update of both PF and KF. Therefore, the quantities computed for the measurement update of the PF can be reused for the KF measurement update. In particular, each particle is only evaluated once in line 1 of each MPF iteration instead of twice as with the previous formulation in Reference [46].

### 2.3. Nonlinear Recursive B-Spline Approximation

The Nonlinear recursive B-spline approximation (NRBA) iteratively adapts a B-spline function $f(s)$ with degree $d$ to the data set from Section 1.5. Algorithm 2 states the instructions for one iteration of NRBA, which is based on the MPF.

In each iteration $t$, NRBA modifies $f$ in $I \in \mathbb{N}$ consecutive spline intervals. Each linear particle $\hat{x}_t^{L,p} = (\hat{x}_{t_1}^L, \hat{x}_{t_2}^L, \ldots, \hat{x}_{t_J}^L)^\top$ and each nonlinear particle $\hat{x}_t^{N,p} = (\hat{x}_{t_1}^N, \hat{x}_{t_2}^N, \ldots, \hat{x}_{t_J}^N)^\top$ contains estimates for $J = d + I$ function coefficients of $f$. $\kappa_t = (\kappa_{t_1}, \kappa_{t_2}, \ldots, \kappa_{t_K})$ denotes the knot vector comprising $K = J + d + 1$ knots. The resulting definition range $\mathcal{D}_t$ of $f$ is given by $\mathcal{D}_t = [\kappa_{t_{d+1}}, \kappa_{t_{J+1}})$. NRBA checks if $s_t$ is in the definition range of the previous time step, $\mathcal{D}_{t-1}$. If not, $\mathcal{D}_{t-1}$ needs to be shifted such that $s_t \in \mathcal{D}_t$. A shift can be conducted in the MPF time update. The result of the time update is the a priori estimate $\hat{x}_t^-$. In the following measurement update, we need $s_t$ again to compute the measurement matrix $\mathcal{C}_t$, and then, to take into account $y_t$. The result of the measurement update is the a posteriori estimate $\hat{x}_t^+$.

Figure 1 depicts the allocation of available data points and computed estimates $\hat{x}$ to KF iterations in RBA versus MPF iterations in NRBA. The arrows indicate the needed information for computing the estimates. The KF is initialized with $\hat{x}_0^+$ and conducts in each iteration a time update first and then a measurement update. Therefore, we need $n$ iterations for $n$ data points. In contrast, the MPF performs the measurement update first and is initialized with $\hat{x}_0^-$. Therefore, we have to save $y_t$ and provide $s_t$, $s_{t+1}$, and $y_t$ for iteration $t + 1$. Hence, we need one iteration more than with the KF in order to take into account all data points. By definition, we use $(s_1, y_1)$ for computing $\hat{x}_0^+$ and $s_n$ for $\hat{x}_{n+1}^-$ as indicated by the dashed arrows.

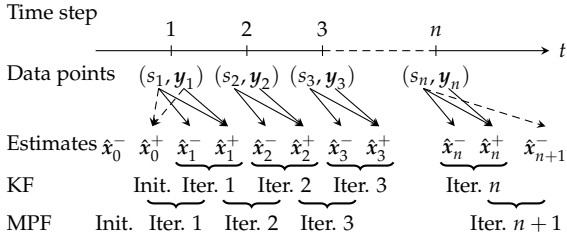

**Figure 1.** The allocation of the available data points and computed estimates $\hat{x}$ to KF iterations in RBA versus MPF iterations in NRBA: The arrows indicate the needed information for computing the estimates. By definition, we use $(s_1, y_1)$ for computing $\hat{x}_0^+$ and $s_n$ for $\hat{x}_{n+1}^-$ as indicated by the dashed arrows.

#### 2.3.1. Initialization

Each linear particle $\hat{x}_0^{L,-,p}$ is initialized with $\hat{x}_0^{L,-,p} = \bar{x}^{\text{Init}} \mathbf{1}_{J \times 1}$, and each nonlinear particle $\hat{x}_0^{N,-,p}$ is initialized with $\hat{x}_0^{N,-,p} = \bar{x}^{\text{Init}} \mathbf{1}_{J \times 1} + \text{chol}\left(\bar{p} I_{J \times J}\right) \cdot \mathbf{rnd}_{J \times 1}$. Hereby, $\mathbf{1}_{J \times 1}$ is a $J \times 1$ matrix of ones and $\bar{x}^{\text{Init}}$ indicates an initial value equal to the scalar measurement $y_{1,v}$ referring to $f$. $\text{chol}(\cdot)$ computes the Cholesky factorization, and $\mathbf{rnd}_{J \times 1}$ is a $J \times 1$ vector of random values drawn from the standard normal

distribution. The covariance matrix of a priori estimation error of linear states, $\mathcal{P}^{L,-}$, is initialized with $\mathcal{P}_0^{L,-} = \bar{p} I_{J \times J}$. $I_{J \times J}$ denotes a $J \times J$ identity matrix.

---

**Algorithm 2:** Nonlinear recursive B-spline approximation

**Input:** $\kappa_{t-1}, \hat{x}_{t-1}^{L,-,p}, \hat{x}_{t-1}^{N,-,p}, \hat{x}_{t-2}^+, \mathcal{P}_{t-1}^{L,-}, \mathcal{R}_{t-1}, s_t, s_{t-1}, y_t, y_{t-1}, \bar{\kappa}_t, \bar{p}, \bar{q}^L, \bar{q}^N, c$

1   $J \leftarrow$ **length** $\left( \hat{x}_{t-1}^{L,+,p} \right)$

2   $K \leftarrow$ **length** $(\kappa_{t-1})$

3   $d \leftarrow K - J - 1$

4   $I \leftarrow J - d$

    /* Quantities for MPF measurement update                                                       */

5   Compute $\mu$ such that $s_{t-1} \in \left[ \kappa_{t-1_\mu}, \kappa_{t-1_{\mu+1}} \right)$

6   $V_{t-1} \leftarrow$ **length** $(y_{t-1})$

7   $\mathcal{C}_{t-1} \in \mathbb{R}^{V_{t-1} \times J}$ from (16)

    /* Quantities for MPF time update                                                                 */

8   $\sigma \leftarrow 0$

9   **if** $s_t \geq \kappa_{t-1_{J+1}}$ **then**

10     |   **if** $s_t \geq \kappa_{t-1_K}$ **then**

11     |   |   $\sigma \leftarrow d + 1$

12     |   **else**

13     |   |   Compute $\sigma$ such that $s_t \in \left[ \kappa_{t-1_{d+I+1+\sigma}}, \kappa_{t-1_{d+I+2+\sigma}} \right)$

14     |   **end**

15   **else if** $s_t < \kappa_{t-1_{d+1}}$ **then**

16     |   **if** $s_t < \kappa_{t-1_1}$ **then**

17     |   |   $\sigma \leftarrow -(d+1)$

18     |   **else**

19     |   |   Compute $\sigma$ such that $s_t \in \left[ \kappa_{t-1_{d+1+\sigma}}, \kappa_{t-1_{d+2+\sigma}} \right)$

20     |   **end**

21   **end**

22   **if** $\sigma \geq 0$ **then**

23     |   $\bar{x} \leftarrow$ last element of $\hat{x}_{t-2}^+$

24     |   $\kappa_t, u_t^L, u_t^N$ from (17), (20) and (26)

25   **else**

26     |   $\bar{x} \leftarrow$ first element of $\hat{x}_{t-2}^+$

27     |   $\kappa_t, u_t^L, u_t^N$ from (17), (29) and (30)

28   **end**

29   Compute $\mu$ such that $s_t \in \left[ \kappa_{t_\mu}, \kappa_{t_{\mu+1}} \right)$

30   $\mathcal{A}_t^L, \mathcal{Q}_t^L, \mathcal{A}_t^N$ and $\mathcal{Q}_t^N$ from (18), (23), (25) and (27)

31   $\left[ \mathcal{P}_t^{L,-}, \hat{x}_{t-1}^+, \hat{x}_t^{L,-,p}, \hat{x}_t^{N,-,p} \right] \leftarrow$
      Algorithm 1 $\left( \mathcal{A}_t^L, \mathcal{A}_t^N, \mathcal{C}_{t-1}, c, \mathcal{P}_{t-1}^{L,-}, \mathcal{Q}_t^L, \mathcal{Q}_t^N, \mathcal{R}_{t-1}, u_t^L, u_t^N, \hat{x}_{t-1}^{L,-,p}, \hat{x}_{t-1}^{N,-,p}, y_{t-1} \right)$

**Output:** $\kappa_t, \hat{x}_{t-1}^+, \hat{x}_t^{L,-,p}, \hat{x}_t^{N,-,p}, \mathcal{P}_t^{L,-}$

---

The large scalar value $\bar{p}$ causes $\hat{x}_t$ to quickly change such that $f$ adapts to the data. Provided that the values in $\mathcal{Q}_t^L$ are small, the values in $\mathcal{P}_t^{L,-}$ decrease as $t$ grows because of line 8 of Algorithm 1. Small elements in $\mathcal{P}_t^{L,-}$ correspond to certain estimates. Therefore, the particles $\hat{x}_t^{L,-,p}$ and $\hat{x}_t^{N,-,p}$ are slower to be updated using measurements such that $f$ converges. Analogous statements hold for $\mathcal{P}_t^{\xi,-}$ because of line 9 of Algorithm 1.

Hence, the process noises are defined as $\mathcal{Q}_t^L = \bar{q}^L I_{J \times J}$ and $\mathcal{Q}_t^N = \bar{q}^N I_{J \times J}$ with small positive values $\bar{q}^L$ and $\bar{q}^N$, respectively.

### 2.3.2. Measurement Update

The measurement update from line 1 to line 4 of Algorithm 1 adapts $f(s)$ based on $(s_{t-1}, y_{t-1})$.

The $v$th dimension of $y_{t-1}$ refers to either $f$ itself or a derivative of $f$. Therefore, the $v$th row of the $V_{t-1} \times J$ measurement matrix $\mathcal{C}_{t-1}$ reads

$$\mathcal{C}_{t-1_{v;1,\ldots,J}} = \left( \mathbf{0}_{1 \times (\mu - (d+1))}, \frac{\partial^r}{\partial s^r} \mathbf{b}_{\mu,d}(s_{t-1}), \mathbf{0}_{1 \times (J-\mu)} \right), \tag{16}$$

whereby $s_{t-1} \in [\kappa_\mu, \kappa_{\mu+1})$ and $r \in \mathbb{N}_0$. Algorithm 2 computes $\mathcal{C}_{t-1}$ in line 7 using Equation (16).

The value of the nonlinear measurement function $c$ depends on the nonlinear particles $\hat{x}_{t-1}^{N,-,p}$. Furthermore, $c$ can depend on additional quantities that vary with the application and are not stated in Algorithm 1.

The diagonal $V_t \times V_t$ covariance matrix of measurement noise $\mathcal{R}_{t-1}$ enables a relative weighting of the dimensions of $y_{t-1}$ because the influence of the $v$th dimension of the measurement error $e_t^p = (y_{t-1} - \hat{y}^p)$ on $\hat{x}_{t-1}^{L,-,p}$ and $\hat{x}_{t-1}^{N,-,p}$ decreases with a growing positive value $\mathcal{R}_{t-1_{v;v}}$.

### 2.3.3. Time Update with Shift Operation

Based on a comparison between $\kappa_{t-1}$ and $s_t$, NRBA decides if a shift operation of the B-spline function definition range is required to achieve that $s_t \in \mathcal{D}_t$.

The variable $\sigma$ calculated from line 8 to line 21 of Algorithm 2 states the shift direction of $\mathcal{D}_{t-1}$ and by how many positions components in $\kappa_{t-1}$, $\hat{x}_{t-1}^{L,-,p}$ and $\hat{x}_{t-1}^{N,-,p}$ need to be moved for that purpose. $\sigma > 0$ indicates a right shift of $\mathcal{D}_{t-1}$, $\sigma < 0$ indicates a left shift, and $\sigma = 0$ means that no shift is conducted because $s_t \in \mathcal{D}_{t-1}$.

Algorithm 2 expects that, for $\sigma > 0$, the $|\sigma|$ additionally needed knots are the $\sigma$ last entries of the knot vector $\bar{\kappa}_t = (\bar{\kappa}_{t_1}, \bar{\kappa}_{t_2}, \ldots, \bar{\kappa}_{t_K})$ and that they are the $-\sigma$ first entries of $\bar{\kappa}_t$ if $\sigma < 0$.

Case 1: Right shift of definition range ($\sigma \geq 0$)

The updated knot vector reads

$$\kappa_t \leftarrow \big( \kappa_{t-1_{\sigma+1}}, \kappa_{t-1_{\sigma+2}}, \ldots, \kappa_{t-1_K}, \tag{17}$$
$$\bar{\kappa}_{t_{K-\sigma+1}}, \bar{\kappa}_{t_{K-\sigma+2}}, \ldots, \bar{\kappa}_{t_K} \big)$$

and line 6 of Algorithm 1 updates $\hat{x}_{t-1}^{L,+,p}$ to $\hat{x}_t^{L,-,p}$ using the state transition matrix

$$\mathcal{A}_t^L = \mathcal{A}_t \tag{18}$$

with

$$\mathcal{A}_t \in \mathbb{R}^{J \times J} \text{ with } \mathcal{A}_{t_{g;h}} = \begin{cases} 1, & \text{if } h = g + \sigma \\ 0, & \text{otherwise.} \end{cases} \tag{19}$$

and the input signal vector

$$u_t^L = u_t \tag{20}$$

with

$$u_t = \left( \mathbf{0}_{1 \times (J - \sigma)}, \bar{x} \mathbf{1}_{1 \times \sigma} \right)^\top . \tag{21}$$

Thereby all entries of $\hat{x}_{t-1}^{L,+,p}$ are moved to the left and the last $\sigma$ entries of $\hat{x}_t^{L,-,p}$ have an arbitrary initial value $\bar{x}$:

$$\hat{x}_t^{L,-,p} = \left( \hat{x}_{t-1_{\sigma+1}}^L, \hat{x}_{t-1_{\sigma+2}}^L, \ldots, \hat{x}_{t-1_{J-\sigma}}^L, \bar{x} \mathbf{1}_{1 \times \sigma} \right)^\top \tag{22}$$

During a right shift of the definition range, we set $\bar{x}$ to the last element of $\hat{x}^+_{t-2}$, which is determined during the preceding call of Algorithm 1 in line 5. This is based on the assumption that $\hat{x}^+_{t-2}$ is a good initial value in the magnitude of the data.

Additionally, line 8 of Algorithm 1 updates $\mathcal{P}^{L,+}_{t-1}$ to $\mathcal{P}^{L,-}_t$ using Equation (18) and

$$\mathcal{Q}^L_t \in \mathbb{R}^{J \times J} \text{ with } \mathcal{Q}^L_{t_{g;h}} = \begin{cases} \bar{p}, & \text{if } h = g \wedge Q \\ \bar{q}^L, & \text{if } h = g \wedge \neg Q \\ 0, & \text{otherwise.} \end{cases} \tag{23}$$

with

$$Q = \begin{cases} h \geq J - \sigma + 1, & \text{if } \sigma \geq 0 \\ h \leq -\sigma, & \text{if } \sigma < 0 \end{cases} \tag{24}$$

The update operation moves the elements in $\mathcal{P}^{L,+}_{t-1}$ to the top left and replaces the zeros on the last $\sigma$ main diagonal elements of $\mathcal{Q}^L_t$ with $\bar{p}$ in order to get large values on the last $\sigma$ main diagonal elements of $\mathcal{P}^{L,-}_t$ and a fast adaption of the initial estimates $\bar{x}$ to the data points.

In line 7 and line 9, Algorithm 1 computes the the quantities $\hat{\xi}^p_t$ and $\mathcal{P}^{\xi,-}_t$ that are needed for the PF time update. The calculations of the state transition matrix $\mathcal{A}^N$ with

$$\mathcal{A}^N_t = \mathcal{A}_t \tag{25}$$

and the input signal vector $u^N$ with

$$u^N_t = u_t \tag{26}$$

are analogous to those for the linear quantities. $\mathcal{Q}^N$ uses $\bar{q}^N$ instead of $\bar{q}^L$:

$$\mathcal{Q}^N_t \in \mathbb{R}^{J \times J} \text{ with } \mathcal{Q}^N_{t_{g;h}} = \begin{cases} \bar{p}, & \text{if } h = g \wedge Q \\ \bar{q}^N, & \text{if } h = g \wedge \neg Q \\ 0, & \text{otherwise.} \end{cases} \tag{27}$$

Case 2: Left shift of definition range ($\sigma < 0$)

The updated knot vector is

$$\kappa_t \leftarrow \left( \bar{\kappa}_{t_1}, \bar{\kappa}_{t_2}, \ldots, \bar{\kappa}_{t_{-\sigma}}, \kappa_{t-1_1}, \kappa_{t-1_2}, \ldots, \kappa_{t-1_{K+\sigma}} \right), \tag{28}$$

the input signal vector for linear states $u^L$ reads

$$u^L_t = u_t \tag{29}$$

and the input signal vector for nonlinear states $u^N$ is given by

$$u^N_t = u_t \tag{30}$$

with

$$u_t \leftarrow \left( \bar{x} \mathbf{1}_{1 \times (-\sigma)}, \mathbf{0}_{1 \times (J+\sigma)} \right)^\top . \tag{31}$$

Additionally, we set $\bar{x}$ to the first component of $\hat{x}^+_{t-2}$.

Note that since $\mathcal{A}^L_t$ and $\mathcal{A}^N_t$ are identical in the chosen state-space model, we can save computational effort when calculating the covariances and cross-covariances from line 8 to line 11 in Algorithm 1.

#### 2.3.4. Effect of the Shift Operation

The shift operation decouples the dimension of the state vector from the total number of estimated coefficients. As a result, NRBA can determine an unknown and unbounded number of coefficients while the effort per iteration only depends on the number of spline intervals in which the approximating function can be adapted simultaneously.

However, the shift operation causes NRBA to partially forget the approximation result in order to keep the dimensions of matrices and vectors constant. $\kappa_t$ and $\hat{x}_t$ only allow an evaluation of $f(s)$ for $s \in [\kappa_{t_{d+1}}, \kappa_{t_{d+I+1}})$. The forgetting mechanism can be circumvented by copying old NRBA elements before they are overwritten.

### 3. Numerical Experiments

We apply Algorithm 2 in numerical experiments. Thereby, we also investigate the effects of the number of simultaneously adaptable spline intervals and the particle count on the NRBA solution. An implementation in MATLAB is provided in [49]. The LM algorithm [50] with MATLAB standard settings serves as a benchmark.

#### 3.1. General Experimental Setup

The data set $\{(s_t, \boldsymbol{y}_t)\}_{t=1,2,\dots,n}$ is defined according to Section 1.5, whereby

$$s_t = 0.25 + 0.5 \cdot (t - 1), \tag{32}$$

$$y_{t,1} = \begin{cases} 40, \text{ if } 80 \le s_t < 120 \\ 30, \text{ otherwise} \end{cases} \tag{33}$$

$$y_{t,2} = y_{t,3} = y_{t,4} = 0 \; \forall t \tag{34}$$

$$n = 400. \tag{35}$$

A B-spline function $f(s)$ of degree $d = 3$ and with knot vector $\kappa = (-30, -20, \dots, 230)$ approximates the data. Thereby, we suppose that $y_{t,1}$ refers to $f$, $y_{t,2}$ to the first derivative of $f$, $y_{t,3}$ to the second derivative of $f$, and $y_{t,4}$ to the value of the nonlinear measurement function $c$.

The nonlinear measurement function $c$ is defined as a quadratic B-spline function with $\kappa = (-5, 0, \dots, 70)$ and $\boldsymbol{x} = (0, 0, 0, 0.25, 1.5, 5, 5, 0, 0, 6, 8, 8, 8)^\top$. $c$ depends on the value of the approximating function $f(s)$ and is displayed in Figure 2. The input variable $f(s)$ of $c$ is restricted to the definition range $[5, 60]$ of $c$.

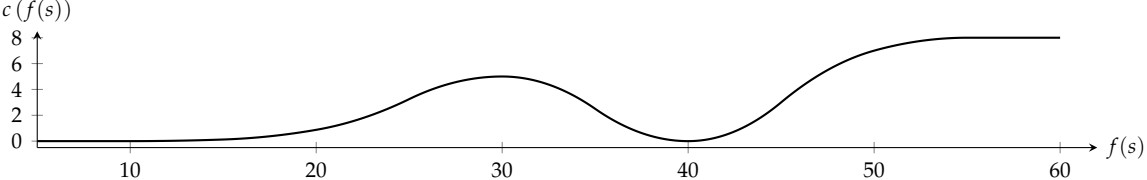

**Figure 2.** The nonlinear measurement function $c(f(s))$ that depends on the value of the B-spline function $f(s)$ that approximates the data. $c$ is itself a B-spline function.

The diagonal measurement covariance matrix $\mathcal{R}_t \in \mathbb{R}^{4\times4}$ with $\mathcal{R}_{t_{1;1}} = 1$, $\mathcal{R}_{t_{2;2}} = 5 \cdot 10^{-2}$, $\mathcal{R}_{t_{3;3}} = 5 \cdot 10^{-3}$ and $\mathcal{R}_{t_{4;4}} = 0.8$ or $10^6$, respectively, comprises the reciprocal weights of $y_{t,1}$, $y_{t,2}$, $y_{t,3}$ and $y_{t,4}$. The reciprocal weight values for the first three dimensions of $\boldsymbol{y}_t$ avoid that $f$ oscillates and cause that $f$ smooths the jumps in the first dimension of the measurements. With $\mathcal{R}_{t_{4;4}} = 0.8$, we weight the nonlinear target criterion $c(f(s)) = 0$ heavily, whereas with $\mathcal{R}_{t_{4;4}} = 10^6$, it is almost completely neglected.

Depending on the applied algorithm, solutions for the former weighting case are denoted with $\text{NRBA}^N$ or $\text{LM}^N$, indicating the nonlinear approximation. Solutions for the latter case are denoted with $\text{NRBA}^L$ or $\text{LM}^L$, indicating that we apply the corresponding algorithm to a quasi-linear approximation problem.

We analyze solutions for two different numbers of spline intervals $I$. For $I = 1$, we initialize $\boldsymbol{\kappa}$ with $\boldsymbol{\kappa}_0 = (-30, 20, \ldots, 40)$, which leads to an initial definition range $[0, 10)$ of $f$. For $I = 3$, we initialize $\boldsymbol{\kappa}$ with $\boldsymbol{\kappa}_0 = (-30, 20, \ldots, 60)$, and the resulting definition range is $[0, 30)$. In both cases, NRBA approximates the data by repeatedly shifting the function definition range to the right. Each time, an additional knot value $\bar{\kappa}_{t_K}$ needs to be provided in the vector $\bar{\boldsymbol{\kappa}}_t$. For $I = 1$, these values are $\bar{\kappa}_{t_K} = 50, 60, \ldots, 230$, and for $I = 3$, they read $\bar{\kappa}_{t_K} = 70, 80, \ldots, 230$.

In order to display the NRBA results for the whole data set, we store all values that are moved out of NRBA matrices and vectors elsewhere. The remaining NRBA parameters are $\bar{q}^L = 0.005$, $\bar{q}^N = 0.25$, and $\bar{p} = 30$. The LM algorithm uses $\bar{x}^{\text{Init}} = 30$ as the initial value for each coefficient.

Due to the included PF, NRBA is a sampling-based, nondeterministic method and its results vary between different approximation runs. Therefore, we apply a Monte Carlo analysis and perform 50 runs for each approximation setting. For each run, we calculate the normalized root mean square error (NRMSE) between the B-spline function determined by NRBA, $f_{\text{NRBA}}$, and the B-spline function according to LM, $f_{\text{LM}}$, as follows:

$$\text{NRMSE} = \frac{1}{\max_{t=1,\ldots,n}\{f_{\text{LM}}(s_t)\} - \min_{t=1,\ldots,n}\{f_{\text{LM}}(s_t)\}} \cdot \sqrt{\frac{\sum_{t=1}^{n} (f_{\text{NRBA}}(s_t) - f_{\text{LM}}(s_t))^2}{n}} \tag{36}$$

With the notation $\text{NRMSE}^{\min}$, $\text{NRMSE}^{\text{med}}$, and $\text{NRMSE}^{\max}$, we refer to the NRBA solution with the minimum, median, or maximum NRMSE, respectively, in each set of 50 runs.

### 3.2. Effect of Weighting and Nonlinear Measurement Function

Figure 3 shows the approximating functions of each algorithm for both $\mathcal{R}_{t_{4;4}} = 0.8$ and $\mathcal{R}_{t_{4;4}} = 10^6$. It displays for each weighting the NRBA solutions that achieve the median and the maximum NRMSE compared to the LM solution with a same weighting. $I$ is set to one for all NRBA approximations; hence, the MPF state vector comprises four linear and four nonlinear components. Furthermore, we choose $P = 6561 = 9^4$; therefore, the PF creates nine samples per nonlinear state dimension.

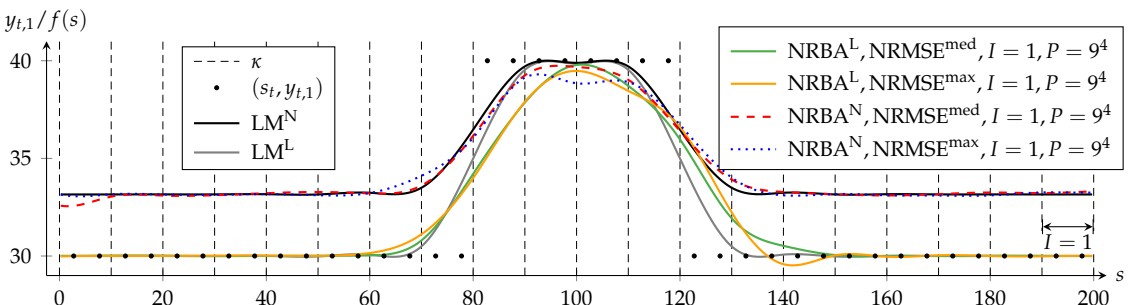

**Figure 3.** Approximating the B-spline function $f$ determined by NRBA with a number of spline intervals $I = 1$ and particle count $P = 6561 = 9^4$ in comparison to the LM solution: $\text{NRBA}^L$ and $\text{LM}^L$ denote solutions of the algorithms for the quasi-linear problem whereas $\text{NRBA}^N$ and $\text{LM}^N$ refer to solutions for the nonlinear problem. $\text{NRMSE}^{\text{med}}$ and $\text{NRMSE}^{\max}$ denote the NRBA solution with the median or maximum normalized root mean square error (NRMSE) compared to the LM solution with the same weighting. Forty of the 400 data points $(s_t, y_{t,1})$ and the knots $\kappa = 0, 5, \ldots, 200$ are shown. The arrow indicates the range in which NRBA can adapt $f(s)$, while the data in the interval $[190, 200)$ is processed.

The black dots depict the first component $y_{t,1}$ of the data points $(s_t, \boldsymbol{y}_t)$. For a better visualization of the approximating functions, only two representative data dots per spline interval are displayed. For $f(s) = 30$, the deviation between the value of $c$ and its target value $y_{t,4} = 0$ has a local maximum (c.f. Figure 2). In NRBA$^N$ and LM$^N$, this deviation is penalized strongly; hence, these solutions avoid $f(s) = 30$. In contrast, NRBA$^L$ and LM$^L$ approximate data with $y_{t,1} = 30$ closely because the nonlinear criterion is weighted only to a negligible extent.

Dashed vertical lines indicate knots, whereby the first and last knots are not shown. Data and knot vector are symmetrical to the straight line defined by $s = 100$. Since the LM algorithm processes all data simultaneously in each iteration, the solutions LM$^L$ and LM$^N$ in Figure 3 reflect this symmetry.

In contrast, NRBA processes the data from left to right and can only adapt some coefficients at a time. For $I = 1$, these are the four coefficients that influence the B-spline function in the spline interval in which the current data point lies. The double-headed arrow in Figure 3 visualizes the range in which NRBA can adapt $f$ simultaneously while $(s_n, \boldsymbol{y}_n)$ is taken into account.

The solutions NRBA$^L$ and NRBA$^N$ are both asymmetrical and mostly delayed with respect to LM$^L$ and LM$^N$. However, with NRBA$^N$, the asymmetry is less distinct. The reason for this is that, in the nonlinear problem, the PF removes states with a high delay more quickly from the particle set because they create a larger error. Additionally, the range of values in NRBA$^N$ is smaller than in NRBA$^L$ so that a present lag is less obvious.

Furthermore, we see that, for the same weighting, NRMSE$^{med}$ and NRMSE$^{max}$ differ only slightly. This suggests that, for the investigated settings, $P = 6561$ suffices for a convergence of NRBA solutions.

### 3.3. Effect of Interval Count

The number of spline intervals $I$ determines the number of intervals in which NRBA can adapt the approximating B-spline function simultaneously.

When we proposed the algorithm RBA for a linear-weighted least squares approximation in Reference [43], we conducted numerical experiments similiar to the ones in this publication but without any nonlinear approximation criterion. For $I = 1$, we observed a strong asymmetry and delay with RBA, analogous to NRBA$^L$ in Figure 3. The filter delay diminished when $I$ was increased to seven. This is because the filter is then able to update more coefficient estimates with hinsight based on $\mathcal{P}^{L,+}$.

In this subsection, we investigate the effect of increasing $I$ from one to three with NRBA. With $I = 3$, NRBA can simultaneously adapt not only the coefficients that are relevant for the spline interval in which the current data point lies but also the two coefficients that affect the two spline intervals to the left. However, $I$ also determines the dimension of the state space. With $I = 3$, there are six linear and six nonlinear components. The PF samples the state space less densely unless the particle count is increased exponentially with $I$.

First, we keep the sampling density per nonlinear state space dimension constant by choosing $P = 625 = 5^4$ for $I = 1$ and $P = 15{,}625 = 5^6$ for $I = 3$.

Figure 4 displays the results for the quasi-linear approximation problem. With $I = 3$, the NRBA solution is more symmetrical than with $I = 1$ for $70 \le s < 120$ as it follows the increase of $y_{t,1}$ more closely. However, a comparison of NRMSE$^{med}$ for $I = 1$ and $I = 3$ indicates that the increase of $I$ does not translate to a reduction of the delay for $s \ge 120$. The ability to adapt more coefficient estimates with hinsight can also lead not necessarily to beneficial effects. The examples are the too low course of NRBA for $I = 3$ between $s = 40$ and $s = 60$ and the overcompensation of the delay between $s = 60$ and $s = 75$.

For $I = 1$, NRMSE$^{max}$ differs more from NRMSE$^{med}$ and shows larger oscillation amplitudes between $s = 130$ and $s = 170$ than for $I = 3$. This suggests that $P = 625$ is not sufficient for a convergence of NRBA for $I = 1$. Although we use only 625 particles for $I = 1$, the required increase to $P = 15{,}625$ for $I = 3$ is quite strong. This illustrates that keeping the sampling density constant quickly becomes infeasible, especially if computation time constraints are present [44]. Figure 5 shows the results for the nonlinear approximation problem and supports the previously drawn conclusions.

Additionally, we see for $s \leq 20$, that the conflicting target criteria in the nonlinear approximation problem cause a larger period for stabilization.

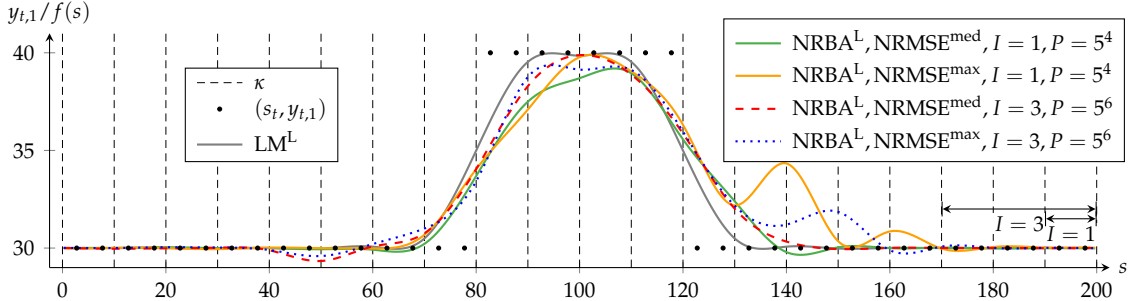

**Figure 4.** Approximating B-spline function, $f$ is determined by NRBA for various numbers of spline intervals $I$ and various particle counts $P$ in comparison to the LM solution. $\text{NRBA}^L$ and $\text{LM}^L$ denote solutions of the corresponding algorithm for the quasi-linear approximation problem. $\text{NRMSE}^{\text{med}}$ and $\text{NRMSE}^{\text{max}}$ denote the NRBA solution with the median or maximum normalized root mean square error (NRMSE) compared to the LM solution with the same weighting. Forty of the 400 data points $(s_t, y_{t,1})$ and the knots $\kappa = 0, 5, \ldots, 200$ are shown. The arrows indicate the range in which NRBA can adapt $f(s)$, while the data in the interval $[190, 200)$ is processed.

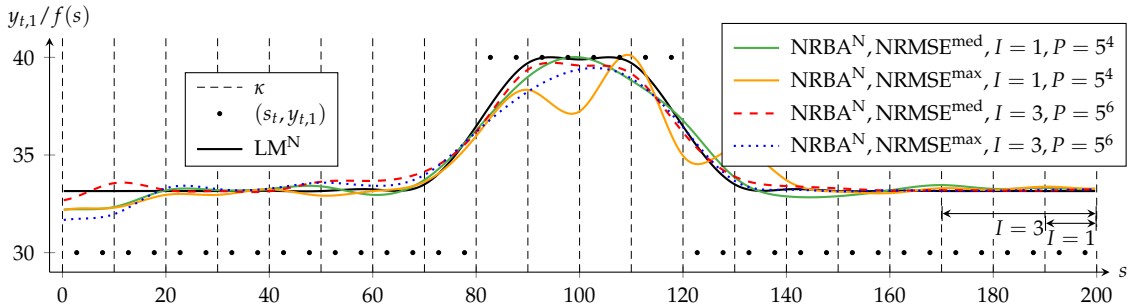

**Figure 5.** Approximating B-spline function, $f$ is determined by NRBA for various numbers of spline intervals $I$ and various particle counts $P$ in comparison to the LM solution. $\text{NRBA}^N$ and $\text{LM}^N$ denote solutions of the corresponding algorithm for the nonlinear approximation problem. $\text{NRMSE}^{\text{med}}$ and $\text{NRMSE}^{\text{max}}$ denote the NRBA solution with the median or maximum normalized root mean square error (NRMSE) compared to the LM solution with the same weighting. Forty of the 400 data points $(s_t, y_{t,1})$ and the knots $\kappa = 0, 5, \ldots, 200$ are shown. The arrows indicate the range in which NRBA can adapt $f(s)$, while the data in the interval $[190, 200)$ is processed.

Second, we investigate the effect of an exclusive $I$ increase from $I = 1$ to $I = 3$ while maintaining the particle count of Section 3.2. Figure 3 then depicts the case for $I = 1$, and Figure 6 depicts the results for $I = 3$. When we compare in both figures the $\text{NRMSE}^{\text{max}}$ solution to the corresponding $\text{NRMSE}^{\text{med}}$ solution, we notice that they differ much more for $I = 3$. This indicates that more particles are needed for convergence for $I = 3$. Especially, we notice that, with $I = 3$, these differences are much larger for $\text{NRBA}^N$ than for $\text{NRBA}^L$.

With the chosen setup, an increasing $I$ yields no clear approximation improvement when we compare corresponding $\text{NRMSE}^{\text{med}}$ solutions in both figures. Figure 6 also shows that $\text{NRBA}^N$ temporarily decreases below $f(s) = 30$, the position of the maximum of $c$ (c.f. Figure 2). This illustrates how the sequential data processing of filter-based methods can lead to solutions that differ from those of a batch method.

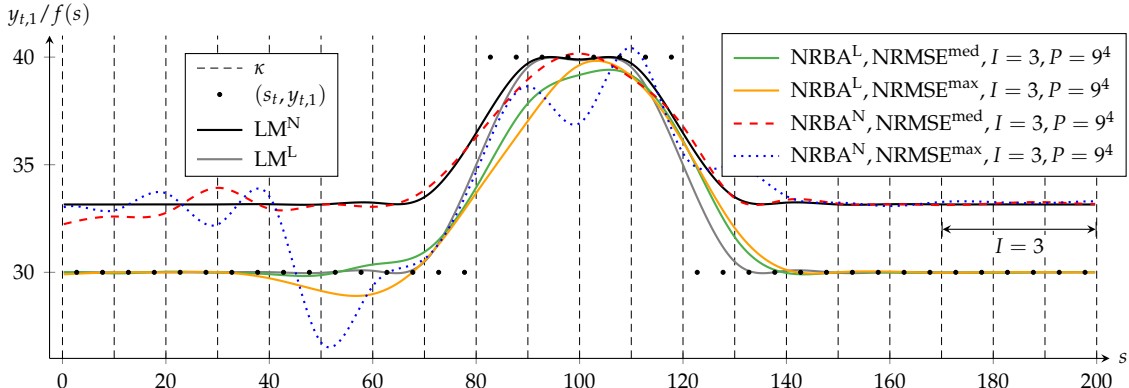

**Figure 6.** Approximating B-spline function, $f$ is determined by NRBA with the number of spline intervals $I = 3$ and particle count $P = 9^4 = 6561$ in comparison to the LM solution. $\text{NRBA}^L$ and $\text{LM}^L$ denote solutions of the algorithms for the quasi-linear problem whereas $\text{NRBA}^N$ and $\text{LM}^N$ refer to solutions for the nonlinear problem. $\text{NRMSE}^{\text{med}}$ and $\text{NRMSE}^{\text{max}}$ denote the NRBA solution with the median or maximum normalized root mean square error (NRMSE) compared to the LM solution with the same weighting. Forty of the 400 data points $(s_t, y_{t,1})$ and the knots $\kappa = 0, 5, \ldots, 200$ are shown. The arrow indicates the range in which NRBA can adapt $f(s)$, while the data in the interval $[190, 200)$ is processed.

### 3.4. Effect of Particle Count on Convergence

The computational effort of MPF increases linearly with the particle count $P$. For an example with seven linear and two nonlinear state vector components, Reference [46] chooses $P = 5000$ and reports that, up to this particle count, increasing $P$ reduces the convergence time significantly and leads to better estimates. Other examples in References [44,51] with four linear and two nonlinear state vector components uses $P = 2000$. The MATLAB example in Reference [47] with three linear components and one nonlinear component uses only $P = 200$.

Figure 7 depicts the effect of $P$ on the convergence of NRBA. For each combination of quasi-linear approximation problem L and nonlinear approximation problem N with $I = 1$ and $I = 3$, the figure shows the courses of $\text{NRMSE}^{\text{min}}$, $\text{NRMSE}^{\text{med}}$, and $\text{NRMSE}^{\text{max}}$ versus $P$. The investigated particle counts are $256 = 4^4, 625 = 5^4, 729 = 3^6, 1296 = 6^4, 2401 = 7^4, 4096 = 4^6 = 8^4, 6561 = 9^4$, and $15,625 = 5^6$.

Between $\text{NRBA}^L$ and $\text{NRBA}^N$, the NRMSE values are on different levels because the LM reference in the NRMSE from Equation (36) differs between $\text{LM}^L$ and $\text{LM}^N$ and the normalization factor in Equation (36) does not fully compensate for this. For $\text{NRBA}^L$ with $I = 1$ and $I = 3$ and $\text{NRBA}^N$ with $I = 1$, $\text{NRMSE}^{\text{min}}$ and $\text{NRMSE}^{\text{med}}$ decrease quickly and remain almost constant when $P$ is further increased from $P = 4096$ on. For $\text{NRBA}^N$ with $I = 3$, the courses of $\text{NRMSE}^{\text{min}}$ and $\text{NRMSE}^{\text{med}}$ suggest using $P = 6561$. $\text{NRMSE}^{\text{max}}$ are the observed worst case results. According to the $\text{NRMSE}^{\text{max}}$ courses, $P = 6561$ should be used for $\text{NRBA}^L$, $P = 15,625$ for $\text{NRBA}^N$ with $I = 1$ and at least $P = 15,625$ for $\text{NRBA}^N$ with $I = 3$.

For $\text{NRBA}^N$ with $I = 3$, $\text{NRMSE}^{\text{max}}$ remains comparatively large because, in some runs, the approximating functions are below $f(s) = 30$ as shown in Figure 6. Only for $P = 15,625$, such results are not observed anymore (c.f. Figure 5) and the $\text{NRMSE}^{\text{max}}$ value is similar to that for $\text{NRBA}^L$. As stated, the heavy penalization of the nonlinear criterion causes the MPF to remove bad particles quickly from the particle set, which reduces the filter lag. However, the MPF then relies more on the state-space sampling on the suboptimal PF than on the optimal KF. In combination with too few particles, this affects the results very negatively in the experiments.

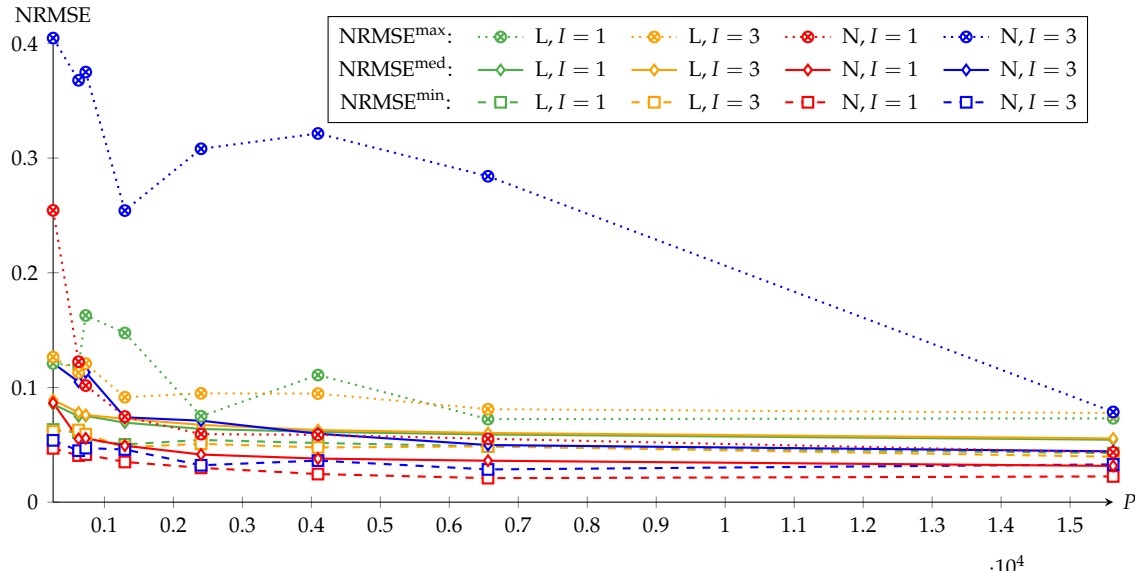

**Figure 7.** The convergence of NRBA: Normalized root mean square error (NRMSE) of NRBA versus the particle count $P$. NRMSE$^{min}$, NRMSE$^{med}$, and NRMSE$^{max}$ denote the nonlinear recursive B-spline approximation (NRBA) solution with the minimum, median, or maximum NRMSE compared to the LM solution in the Monte Carlo analysis. L and N denote the quasi-linear and nonlinear weighting and $I$, the number of spline intervals.

### 3.5. Mean and Standard Deviation of NRBA Error

For insights into the statistical features of NRBA, we determine the mean and standard deviation of the error vector $e$ between the NRBA and LM solutions over all 50 Monte Carlo runs for each approximation setting. Hereby, we consider the error vector of function values between NRBA and LM, $e_f$, as well as the error vector of coefficient values between NRBA and LM, $e_x$. The mean $\bar{e}$ with $\bar{e} = \frac{1}{E}\sum_{i=1}^{E} e_i$ is an indicator for a bias of NRBA estimates, whereas the sample standard deviation $\sigma_e$ with $\sigma_e = \sqrt{\frac{1}{E-1}\sum_{i=1}^{E}(e_i - \bar{e})^2}$ is a measure for bias stability. $e$ is a single error vector component, and $E$ denotes the number of components in the error vector.

Table 1 displays the mean and standard deviation of $e_f$, and Table 2 shows these statistic measures for $e_x$. Both tables enable the following statements:

**Table 1.** The mean and standard deviation of error vector of function values between NRBA and LM over all 50 Monte Carlo runs with a quasi-linear approximation problem $L$, nonlinear approximation problem $N$, and number of spline intervals $I$.

| Particle Count | Mean of Error Vector | | | | Standard Deviation of Error Vector | | | |
|---:|---:|---:|---:|---:|---:|---:|---:|---:|
| | $L, I = 1$ | $L, I = 3$ | $N, I = 1$ | $N, I = 3$ | $L, I = 1$ | $L, I = 3$ | $N, I = 1$ | $N, I = 3$ |
| 256 | 0.0041 | $-0.0088$ | $-0.2268$ | $-0.5820$ | 0.8738 | 0.9143 | 0.6225 | 1.2525 |
| 625 | 0.0150 | 0.0072 | $-0.0979$ | $-0.4386$ | 0.7692 | 0.8224 | 0.4030 | 1.0902 |
| 729 | $-0.0064$ | $-0.0176$ | $-0.0930$ | $-0.4350$ | 0.8231 | 0.7988 | 0.3904 | 1.0975 |
| 1296 | $-0.0028$ | $-0.0156$ | $-0.0611$ | $-0.2248$ | 0.7294 | 0.7365 | 0.3361 | 0.6988 |
| 2401 | 0.0005 | $-0.0009$ | $-0.0445$ | $-0.1851$ | 0.6480 | 0.6851 | 0.2965 | 0.6454 |
| 4096 | 0.0014 | 0.0050 | $-0.0296$ | $-0.2189$ | 0.6436 | 0.6538 | 0.2583 | 0.7106 |
| 6561 | 0.0011 | $-0.0069$ | $-0.0334$ | $-0.1340$ | 0.5930 | 0.6084 | 0.2498 | 0.5673 |
| 15,625 | 0.0056 | $-0.0015$ | $-0.0204$ | $-0.0512$ | 0.5502 | 0.5715 | 0.2216 | 0.3124 |

The mean of the error vector of the quasi-linear approximation problem is not clearly influenced by the particle count. Furthermore, the varying signs of the means close to zero speak against a bias for the quasi-linear approximation problem. In contrast, for the nonlinear approximation problem,

the means are always negative and, therefore, biased. The negative sign is a problem-specific result and means that the solution NRBA$^N$ is, in general, between LM$^L$ and LM$^N$. The bias itself, however, seems to be a systematic effect of the interaction of KF and PF of which the system models are weighted relative to each other according to the covariance matrix of process noise for linear states $\mathcal{Q}^L$ and the covariance matrix of process noise for nonlinear states $\mathcal{Q}^N$. Decreasing $\mathcal{Q}^N$ might help to reduce the magnitude of this bias. Moreover, we note that the absolute values of the means become smaller as the particle count is increased. As the NRBA results for nonlinear approximation problem rely heavily on the PF, this relation is comprehensible.

**Table 2.** The mean and standard deviation of error vector of coefficient values between NRBA and LM over all 50 Monte Carlo runs with a quasi-linear approximation problem $L$, a nonlinear approximation problem $N$, and number of spline intervals $I$.

| Particle Count | Mean of Error Vector | | | | Standard Deviation of Error Vector | | | |
|---|---|---|---|---|---|---|---|---|
| | $L, I = 1$ | $L, I = 3$ | $N, I = 1$ | $N, I = 3$ | $L, I = 1$ | $L, I = 3$ | $N, I = 1$ | $N, I = 3$ |
| 256 | 0.0041 | −0.0098 | −0.3036 | −0.5902 | 1.2019 | 1.3057 | 1.2664 | 1.7251 |
| 625 | 0.0133 | 0.0044 | −0.1640 | −0.4562 | 1.0725 | 1.1953 | 0.8450 | 1.4869 |
| 729 | −0.0040 | −0.0055 | −0.1507 | −0.4742 | 1.1971 | 1.1239 | 0.8092 | 1.5128 |
| 1296 | −0.0024 | −0.0171 | −0.1212 | −0.2695 | 1.0045 | 1.0828 | 0.7122 | 1.1541 |
| 2401 | 0.0005 | 0.0001 | −0.1099 | −0.2271 | 0.9309 | 0.9916 | 0.6654 | 1.0487 |
| 4096 | 0.0019 | 0.0072 | −0.0725 | −0.2511 | 0.9334 | 0.9402 | 0.5621 | 1.0439 |
| 6561 | 0.0013 | −0.0053 | −0.0738 | −0.1889 | 0.8614 | 0.9000 | 0.5201 | 0.9672 |
| 15,625 | 0.0054 | −0.0008 | −0.0506 | −0.1064 | 0.8211 | 0.8329 | 0.4605 | 0.6524 |

The standard deviation, in general, decreases for all investigated settings as the particle count is increased. With 15,625 particles, the standard deviations of the quasi-linear approximation problems are two to three times larger than those of the nonlinear approximation problems. This also is a problem-specific effect. For the nonlinear approximation problem, the range of the function values is considerably smaller than that of the quasi-linear problem, which favors lower standard deviations. For the nonlinear problem with the number of spline intervals equal to three, the relatively large standard deviations reflect the often disadvantageous courses of the approximating functions again as depicted in Figure 6.

## 4. Trajectory Optimization

This section demonstrates how NRBA can be applied for a multiobjective trajectory optimization. The trajectory represents the planned vehicle velocity with respect to time $\tau$ measured from present into the future and is a B-spline function as defined in Equation (4) with degree $d = 3$, knot vector $\kappa$, and coefficient vector $x$. Due to its interpretation as a temporal velocity trajectory, we refer to the B-spline function as $v_{TJY}(\tau)$ instead of $f(s)$. $\kappa$ has equidistant and strictly monotonously increasing entries

$$\kappa = (\kappa_1, \kappa_2, \ldots, \kappa_K) = (-\Delta\tau_\kappa \cdot d, \Delta\tau_\kappa \cdot (d+1), \ldots, \Delta\tau_\kappa \cdot d + K - 1) \tag{37}$$

where $\Delta\tau_\kappa$ denotes the constant temporal distance of neighboring knots. Due to the choice of $\kappa$, $v_{TJY}(\tau)$ can be evaluated for $\tau \geq 0$. $\tau$ is discretized using a positive constant temporal distance of neighboring data points $\Delta\tau_{It}$:

$$\tau_t = (t - 1) \cdot \Delta\tau_{It}, \ t = 1, 2, \ldots, n \tag{38}$$

Each component of the vector of measurements $y_t$ of the data set in Section 1.5 is interpreted as a target value of an optimization goal. $y_{t,1}$ is assumed to be a suggested time-discrete course of velocity with a velocity set point $v_{Set}$ which comes from a preceding planning method:

$$y_{t,1} = v_{\text{Set},t} \tag{39}$$

The remaining components $y_{t,2}$, $y_{t,3}$, and $y_{t,4}$ of $\boldsymbol{y}_t$ are assumed zero as before. NRBA solves the optimization problem

$$\hat{\boldsymbol{x}} = \underset{\boldsymbol{x}}{\arg\min} \sum_{t=1}^{n} \left( R_v^{-1} \cdot (v_{\text{Set},t} - v_{\text{TJY}}(\tau_t))^2 + R_a^{-1} \cdot a_{\text{TJY}}(\tau_t)^2 + R_j^{-1} \cdot j_{\text{TJY}}(\tau_t)^2 + R_{\text{P}}^{-1} \cdot \hat{P}_{\text{elec}}(\tau_t)^2 \right) \tag{40}$$

Each summand of the optimization function refers to an optimization goal. Under the assumption that $v_{\text{Set}}$ takes into account driving dynamics, the first summand can be interpreted as driving safety and the optimized trajectory should remain close to the course of $v_{\text{Set}}$. $a_{\text{TJY}}$ denotes the trajectory acceleration and $j_{\text{TJY}}$ the trajectory jerk. These quantities are the first and second derivatives of $v_{\text{TJY}}$ and can be derived according to Equation (6). The second and third summands demand a smooth drive with a low acceleration and acceleration changes and, thus, refer to driving comfort. The last summand penalizes the absolute values of the estimated electric traction power $\hat{P}_{\text{elec}}$, which is used as a measure for driving efficiency.

Each optimization goal has a corresponding weight. $R_v^{-1}$ denotes the weight of velocity error square, $R_a^{-1}$ denotes the weight of acceleration error square, $R_j^{-1}$ denotes the weight of jerk error square, and $R_{\text{P}}^{-1}$ denotes the weight of power error square. The reciprocals of the weights follow the interpretation of the filter algorithms and refer to the variances of the artificial measurements. $R_v$ is the variance of velocity measurement, $R_a$ is the variance of acceleration measurement, $R_j$ is the variance of jerk measurement, and $R_{\text{P}}$ is the variance of power measurement.

Without the fourth goal, RBA would suffice for solving the problem because the first three goals are all linear in the coefficients. However, the energy consumption minimization goal requires a nonlinear method. In the following, we consider a BEV based on the Porsche Boxster (type 981), which is described in References [52–54]. Like most BEVs, its powertrain has a fixed gear ratio, which simplifies the optimization problem and allows us to apply NRBA.

In a BEV, the powertrain converts electric traction power $P_{\text{elec}}$ provided by the battery into mechanic traction power $P_{\text{mech}}$ for vehicle propulsion. During recuperative braking, the power flow is vice versa. We will neglect the additional power for auxillaries such as air conditioning because it depends on environmental conditions and comfort requirements strongly. $P_{\text{mech}}$ equals the product of the traction force $F_{\text{trac}}$ and the vehicle velocity $v_{\text{vhcl}}$, whereby $F_{\text{trac}}$ equals the sum of driving resistances. The dominant driving resistances are air resistance. which increases quadratically with $v_{\text{vhcl}}$, the inertial force which is a linear function of the vehicle acceleration $a_{\text{vhcl}}$ and the climbing force which depends on the road slope angle $\alpha$.

During this power conversion, losses occur in various components of the powertrain. In order to provide sufficient $F_{\text{trac}}$ for a high acceleration or high velocity, the electric motor must generate a high torque which requires a large electric current I. The internal ohmic resistance R of electric components such as the battery causes an ohmic traction power loss $P_{\text{loss,ohmic}}$ which is given by $P_{\text{loss,ohmic}} = R \cdot I^2$. Furthermore, friction losses in the gearbox increase with rotation speed and transmitted torque [55].

$P_{\text{elec}}$ can be computed in the vehicle from voltage and current sensor data. However, due to a lack of sensors, $F_{\text{trac}}$ and $P_{\text{mech}}$ cannot be calculated, and therefore, power losses cannot be determined in the vehicle during its operation. As power losses increase with the absolute value of $P_{\text{elec}}$, we use $P_{\text{elec}}$ as a measure for power losses and create a mathematical model of $P_{\text{elec}}$ that outputs the estimated electric traction power $\hat{P}_{\text{elec}}$ based on the inputs $v_{\text{vhcl}}$, $a_{\text{vhcl}}$, and $\alpha$. The mathematical model can adapt its parameters during vehicle operation because both model outputs and model inputs are known quantities during vehicle operation. The adaption is neccessary for accurate estimates because vehicle parameters such as mass or air drag coefficient can change and the driving resistances also depend on these parameters. The mathematical model serves as nonlinear measurement function for NRBA, whereby we assume that $v_{\text{TJY}} = v_{\text{vhcl}}$ and $a_{\text{TJY}} = a_{\text{vhcl}}$. By penalizing the absolute value of $\hat{P}_{\text{elec}}$ in Equation (40), we encourage NRBA to determine energy-efficient velocity trajectories.

The first diagram of Figure 8 displays the velocity $v$ versus the time $\tau$ according to the velocity set points $v_{\text{Set}}$ of the reference as well as three trajectories optimized by NRBA. The NRBA trajectories are denoted NRBA$^1$, NRBA$^2$, and NRBA$^3$ and differ in the choice of $R_{\text{P}}$. We use $R_{\text{P}} = 10^4$ for NRBA$^1$, $R_{\text{P}} = 500$ for NRBA$^2$, and $R_{\text{P}} = 100$ for NRBA$^3$. The remaining parameter values are $R_v = 5$, $R_a = 10$, $R_j = 1$, $I = 1$, $\bar{q}^L = 0.005$, $\bar{q}^N = 0.5$, $\bar{p} = 15$, $P = 1000$, $\Delta\tau_\kappa = 2$, and $\Delta\tau_{\text{It}} = 0.25$.

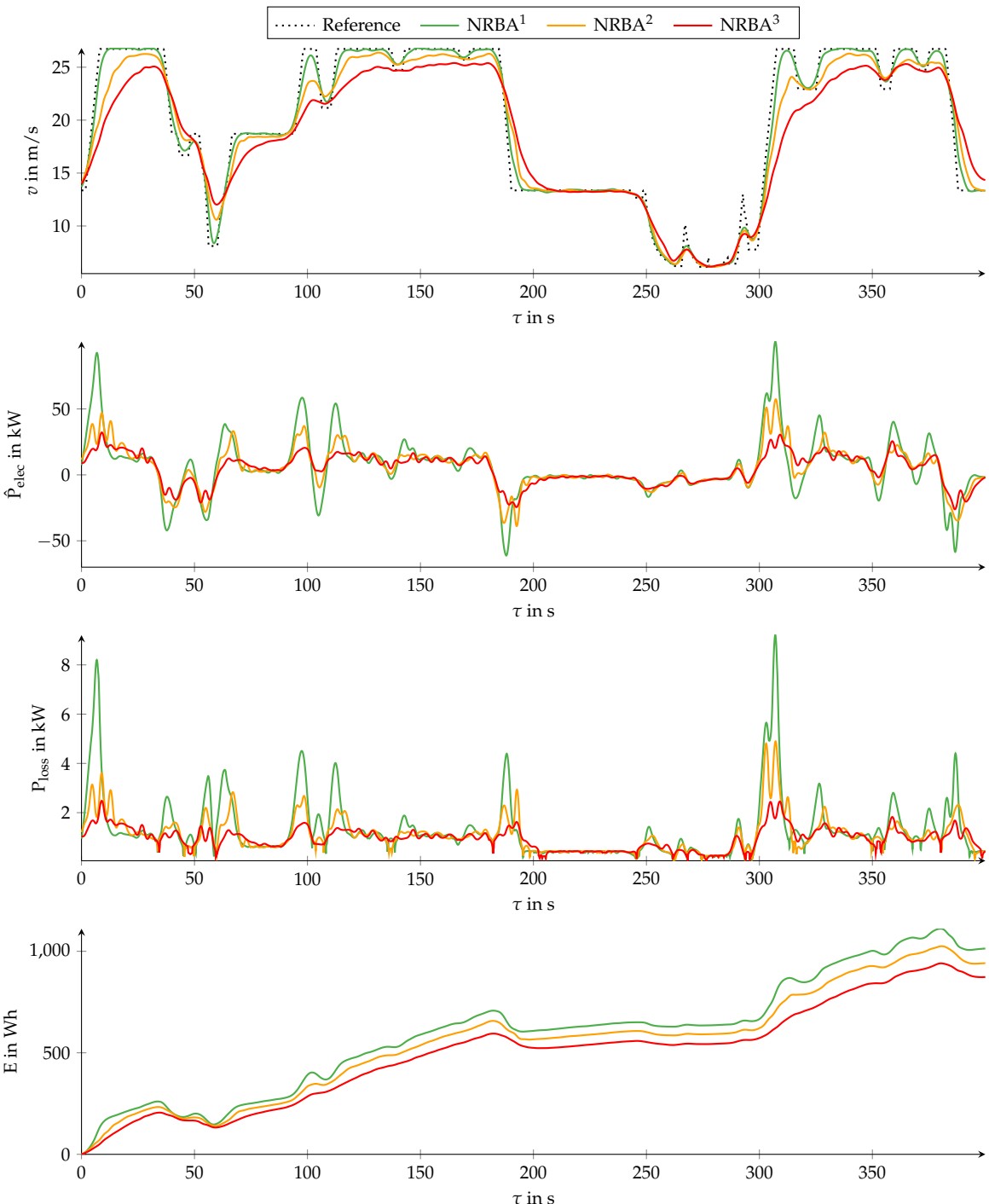

**Figure 8.** First diagram: Velocity $v$ versus time $\tau$ according to the velocity set points $v_{\text{Set}}$ of the reference and three trajectories NRBA$^1$, NRBA$^2$, and NRBA$^3$ optimized by NRBA that differ in the variance of power measurement. Second diagram: Estimated electric traction power $\hat{P}_{\text{elec}}$ according to mathematical model. Third diagram: Traction power loss $P_{\text{loss}}$. Fourth diagram: Traction energy E.

The second diagram shows the estimated electric traction power $\hat{P}_{\text{elec}}$ according to the mathematical model. The traction power loss $P_{\text{loss}}$ and traction energy E depicted in the third and fourth diagram originate from a detailed vehicle model. The detailed vehicle model includes parameters for all relevant power losses. These parameters were derived from component tests on test benches. The detailed model requires $F_{\text{trac}}$ as an input and, therefore, assumptions concerning the driving resistance parameters. For simplicity, we assume a slope-free road in this example. An implementation of this example in MATLAB is also provided in Reference [49].

The trajectory NRBA[1] follows the reference closely apart from some short and large changes of $v_{\text{Set}}$ between $\tau = 250$ and $\tau = 300$. Staying close to the reference requires several positive and negative peaks in $\hat{P}_{\text{elec}}$, which are almost not penalized because of the high variance of power measurement $R_{\text{P}}$. As $R_{\text{P}}$ is decreased, the trajectories exhibit lower velocities and absolute values of acceleration in order to avoid large absolute values of $\hat{P}_{\text{elec}}$. Between $\tau = 250$ and $\tau = 300$ decreasing, $R_{\text{P}}$ has almost no effect because $\hat{P}_{\text{elec}}$ is close to zero because the velocity is low.

The last three diagrams show that $\hat{P}_{\text{elec}}$ is a suitable measure for the goal of a lower energy consumption. A comparison of the peaks in $\hat{P}_{\text{elec}}$ and $P_{\text{loss}}$ at $\tau = 310$ illustrates that $P_{\text{loss}}$ increases with $|\hat{P}_{\text{elec}}|$ more than linearly.

Note that there are some situations in which the trajectories exceed $v_{\text{Set}}$. Depending on the exact application, interpreting $v_{\text{Set}}$ as an upper velocity limit might be more suitable. By penalizing positive deviations $(v_{\text{Set},t} - v_{\text{TJY}}(\tau_t))$ more strongly than negative ones in each NRBA iteration using a sign-dependent $R_v$ value, exceeding $v_{\text{Set}}$ can be avoided.

## 5. Conclusions

We presented a filter-based algorithm denoted nonlinear recursive B-spline approximation (NRBA) that determines a B-spline function such that it approximates an unbounded number of data points in the nonlinear weighted least squares sense. NRBA uses a marginalized particle filter (MPF), also denoted a Rao–Blackwellized particle filter, for solving the approximation problem iteratively. In the MPF, a particle filter (PF) takes into account the approximation criteria that relate to the function coefficients in a nonlinear fashion whereas a Kalman filter (KF) solves any linear subproblem optimally. Thus, the particle count in the PF can be reduced.

As the value of the B-spline function and its derivatives depend linearly on the coefficient values, linear approximation criteria will occur in most approximation applications. The MPF accepts the exactly known values of the B-spline function basis functions as an input and does not need to estimate them like many other nonlinear filters do. Therefore, the MPF enables a reduction in the computational effort and an achievement of better results compared to purely nonlinear filters [46].

NRBA can shift estimated coefficients in the MPF state vector which allows an adaptation of the bounded B-spline function definition range during run-time such that, regardless of the initially selected definition range, all data points can be processed. Additionally the shift operation enables a decrease in the dimension of the state vector for less computational effort.

In numerical experiments, we compared NRBA to the Levenberg-Marquardt (LM) algorithm and investigated the effects of NRBA parameters on the approximation result using a Monte Carlo simulation. Provided that the NRBA parameters are chosen appropriately, the NRBA solution is close to the LM solution apart from some filter-typical delay. For a strong weighting of the nonlinear approximation criteria, the result relies more on the state-space sampling of the PF than on the KF. In combination with too few particles, the approximating function tends to oscillate.

NRBA use cases are nonlinear weighted least squares (NWLS) problems in which a linearization of nonlinear criteria is not desired or promising, for example, because of strong nonlinearities. For linear weighted least squares problems, the recursive B-spline approximation (RBA) algorithm proposed in Reference [43] should be used instead of NRBA. RBA is based on the KF, which computes an optimal solution [38]. In contrast, the PF in NRBA causes NRBA to, at best, reach the same approximation quality provided that the particle count is large enough, which requires more computational effort.

Furthermore, with NRBA, the approximation depends more heavily on the parameterization of the underlying filter algorithm than with RBA. An increase of the number of coefficients that NRBA can adapt simultaneously is not as unambiguously beneficial as with RBA and usually needs to be combined with an exponential increase of the particle count in the PF for an improvement of the approximation.

As demonstrated, NRBA is suitable for an unconstrained multiobjective trajectory optimization. Thereby, a major advantage of NRBA is a linear increase of the computational effort with the number of processed data points as opposed to an exponential increase with most other direct trajectory optimization methods. NRBA can also be applied during the processing of discrete signals in a time domain. Then NRBA can provide a sparse, continuous, and smoothed representation of the signals themselves or of their derivatives.

The chosen MPF formulation allows an easy replacement of the standard KF and PF. For example, Reference [56] presents a PF, in which the particles are determined with a particle swarm optimization, and reports that less particles are needed compared with the standard PF. An improvement of the MPF is proposed by Reference [57]. Investigating these algorithms in combination with NRBA can be the subject of further research.

**Author Contributions:** Conceptualization, J.J. and F.B.; methodology, J.J. and F.B.; software, J.J.; investigation, J.J.; writing—original draft preparation, J.J. and F.B.; writing—review and editing, M.F., F.G; supervision, F.G. and M.F.; project administration, F.G.; funding acquisition, M.F.

**Funding:** This research was funded by the German Federal Ministry of Education and Research under grant number 16EMO0071. The APC was funded by the KIT-Publication Fund of the Karlsruhe Institute of Technology.

**Acknowledgments:** We would like to thank the Ing. h.c. F. Porsche AG, who was among the research project partners, for the provision of vehicle data. Furthermore, we appreciate the valuable comments and suggestions of the anonymous reviewers for the improvement of this paper.

**Conflicts of Interest:** The authors declare no conflict of interest.

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
