# Peer review of "An Iterative Method Based on the Marginalized Particle Filter for Nonlinear B-Spline Data Approximation and Trajectory Optimization"

_mathematics, doi:10.3390/math7040355_

Round 1

Reviewer 1 Report

The article concerns B-spline functions and particle filter which can be used to approximation and optimization trajectory. The paper is well written and it contains algorithms and pseudocode for computer programming.

The main model is presented in state-space form what is very important for control problems.

I have one comment: on page 9 above equation (25) there is transition matrix A with index N. What is it? I think that it is transition matrix contains linear and nonlinear transition matrices but I can be wrong.

Finally I suggest that the article should be accepted.

Author Response

Thank you very much for performing the review and for your helpful comments.

Equation (25) contained a typo, which we corrected. Instead of A_t^L = A_t it must read A_t^N = A_t. A_t^N is the state transition matrix for nonlinear states as stated below equation (11): "The superscripts L and N indicate that the corresponding quantity refers to linear or nonlinear state variables, respectively. A_t denotes the state transition matrix,...". Probably the typo was the reason for why it was unclear.

Reviewer 2 Report

This paper proposes a nonlinear recursive B-spline approximation algorithm based on marginalized particle filter. This newly developed algorithm, should decrease the computational effort of the optimization problem. Although the derivation is presented in a straightforward manner, it lacks careful statements about its validity and much is taken for granted. I recommend the authors do a rewrite of the paper, given the following comments:

1. Introduction:

Although the authors provide a long list (Refs. 1-23) of citations going back decades that motivate the development of B-Spline, the use of NWLS approximation problem, the LM algorithm and the extended Kalman filter, etc., The do not consider the previous research about the marginalized particle filter (MPF), especially in the context of trajectory estimation. I suspect that many potential users of the authors' algorithm may also fail to fully appreciate the motivations for using such a sampling-based approach.  Therefore, I believe the paper would benefit from more direct citation, or explicit description, of specific use cases that could benefit, or have benefited, from incorporation of MPF rather than known LM algorithm or the NWLS optimization approach.

2. Methods:

The authors have developed a nice little algorithm that extends the MPF framework smoothly to applications in which B-spline approximation is used to approximate non-linear trajectories. The algorithm is clearly derived and specified. I did not find any errors in the development, which was very straightforward.  But, there are some points in this section not clear for me. For example, this approach has some advantages over the standard Kalman filter used in a previous development of the author [reference 30 of the literature list] but it is not always the case. Especially, there are different tools to deal with non-linearities, as linearization, Unscented transform, ensemble KF etc. Furthermore, the authors do not analyze the convergence of a sampling-based approach such as the particle filtering. Clearly, this algorithm is approximate, but the authors should put more care into describing the approximations and how accurate the derived state vector is. There will also be a discussion on how to use convex optimization for solving the estimation problem.

3. Numerical experiments:

The result presentation is a little bit disconcerting, because the estimates for a single realization are shown. The NRBA approach is a sampling approach. Therefore, a many Monte Carlo runs (not only one realization) would be interesting, especially in showing any biasing or validation of the nonlinear optimization problem (by calculating for example the root mean square error). Furthermore, it is interesting for the readers to compare the results of RBA presented in Paper 30 of the literature list with the results of the NRBA. The numerical experiments are very similar in both papers.

Author Response

Thank you very much for performing the review and for your helpful comments.

We agree with your suggestions and addressed them as follows in the sections Introduction, Numerical experiments and Conclusions.

In the introduction we added a subsection "Bayesian filters" in which we state the differences of common filters for linear and nonlinear state estimation in order to give some insights into the different ways of dealing with nonlinearities, the approximations used and the advantages and disadvantages of the approaches. Thereby we cite trajectory estimation and optimization applications of the filters.

At the end of this subsection we especially mention the features of the marginalized particle filter (MPF) and reference automotive trajectory optimization applications that benefitted from this filter.

Moreover, we added a sentence in the contribution subsection that states specifically for the proposed algorithm NRBA the advantages of using the MPF.

For the numerical experiments section we performed a Monte Carlo analysis with 50 runs for each setting in order to derive statements regarding the convergence of the proposed sampling-based method NRBA and the required particle count. Reworked figures show the trajectories that achieved the median and maximum of the normalized root mean squared error (NRMSE) compared to  Levenberg-Marquardt reference instead of the results of a single realization. We also added a figure that depicts the courses of minimum, median and maximum NRMSE versus the particle count for different approximation settings. 

The previously submitted manuscript already contains some comparing statements between NRBA and the RBA algorithm for linear weighted least squares approximation in its subsection 3.3 (Effect of interval count) and the conclusions section. In subsection 3.3 we made the statements more precise and mention the results with RBA for the analogous experiments. In the conclusions section we added a paragraph that explicitly mentions use-cases for NRBA as well as cases in which RBA should be preferred instead of NRBA.

Reviewer 3 Report

The presented paper is, in my opinion, original and worth publishing. The authors introduced a new algorithm for nonlinear optimisation. Application of proposed concepts is well illustrated with examples.

I would like to point out, that the NRBA source code given in Ref.[36] is unavailable. By clicking the address 10.5281/zenodo.2532226 you can only find the LICENSE and README.md files. In fact, after some search I found (seemed to be correct) files here 10.5281/zenodo.1346749. However, in the readme file, I found that materials are prepared for a different publication:

[1]: Jens Jauch, Felix Bleimund, Michael Frey, Frank Gauterin: 

%      Nonlinear recursive B-spline approximation using the marginalized particle filter, 

%      Engineering Science and Technology, an International Journal (2018).

Please check both addresses and correct descriptions on the page and provide the correct link in the paper.

Author Response

Thank you very much for performing the review and for your helpful comments.

The doi 10.5281/zenodo.2532226 stated in the manuscript references the initial commit of the github repository in which the actual files were not yet contained.

The doi 10.5281/zenodo.1346749 you found then stems indeed from an earlier preparation of the material for publication in a different journal that rejected the manuscript.

Meanwhile the trajectory optimization example and the Monte Carlo analysis have been added to the Matlab files. We corrected the doi and additionally provide the link to the github repository itself in the manuscript to make sure that the latest files can be found.

Round 2

Reviewer 2 Report

The authors extended in their revised paper all suggested ideas and comments. The authors present their findings in terms of very specific and easily identifiable facts, which I believe would be useful for any who applies their MPF algorithm. The authors have also provided their Matlab code which can be used for any B-spline approximation problem recursively. The methodological/mathematical subject matter was generally in the first version easy to follow and understand. In my view, the extension of the additional subsection (the Bayes filter) in the introduction with a good selection of references is relevant and sufficient. The extensive Monte Carlo simulation provided in the revised version shows the benefits of the new developed approach as well as the analysis of convergence behaviour. I think the Monte Carlo simulation should be extended to estimate the standard error of all Monte Carlo runs. Beside the standard error of the estimated parameter (and the approximated curves), the Bias of the estimate (derived from the Monte Carlo runs) can give some additional information about the statistical accuracy of an estimated parameters and estimated curves as well. I completely missed such an uncertainty analysis in this paper.  I think it will be suitable for publication after considering these minor issues.

Author Response

We addressed your appreciated comments in a new subsection "Mean and standard deviation of NRBA error". This subsection includes two tables, which state the means and standard deviations of both the coefficient vectors determined by NRBA and the resulting approximating functions over all Monte Carlo simulations for the previously investigated approximation settings as suggested. Based on these statistical measures we derive statements on bias and accuracy of NRBA.